# Influence of Different Proportion Intercropping on Oat and Common Vetch Yields and Nutritional Composition at Different Growth Stages

Jiahui Qu, Lijun Li *, Jianhui Bai *, Guangmin Chen, Yanli Zhang and Qing Chang

College of Agronomy, Inner Mongolia Agricultural University, No.275, Xin Jian East Street, Hohhot 010019, China
* Correspondence: imaullj@163.com (L.L.); bzmayyb@163.com (J.B.); Tel.: +86-15848154170 (L.L.)

**Abstract:** Crop yield mainly depends on environment and cultivation practices that vary according to a growing environment. However, an oat (*Avena sativa* L.)-common vetch (*Vicia sativa* L.) intercrop system has not been fully developed in the agro-pastoral ecotone of Inner Mongolia, China. This study evaluated the effects of seven treatments, including five oat-common vetch intercropping patterns, sole oat, and sole vetch on yield and quality performance at different growth periods [75 days after sowing (DAS), 90 DAS, 105 DAS], on the basis of field experiments conducted in the agro-pastoral ecotone of Inner Mongolia in 2015 and 2016. The OV3:1 (oat intercropped with common vetch at seeding ratios 3:1) treatment at 105 DAS in 2016 was superior to other treatments, as it achieved the highest shoot dry matter, increasing by 24.1% and 37.1% compared to sole oat and common vetch. The crude fat (CF) contents, CF yield, and crude protein (CP) yield increased under OV3:1, and acid detergent fiber (ADF) decreased under OV3:1, compared to monoculture. The results indicate that intercropping is an efficient cropping system for the agro-pastoral ecotone of Inner Mongolia. The appropriate proportion of oat and common vetch intercropping at 3:1 and harvesting time not only increases crop yield but also improves the crop quality.

**Keywords:** intercropping; oat; common vetch; yield





## 1. Introduction

The grassland, which has been estimated to be $4 \times 10^8$ hm$^2$ in China, plays a crucial role in national ecological security and food security [1]. However, the output of grass products cannot meet the market demand of animal husbandry development due to a series of problems, such as the water resources shortage, grassland degradation, insufficient investment in the forage industry, extensive production mode and small-scale production of the grassland industry [2]. In addition, great changes have taken place in the dietary structure of urban and rural residents in China, with a decrease in the consumption of rations and an increase in the consumption of animal foods such as meat, eggs and milk, accompanied by a sharp increase in animal-derived protein demand [3,4]. However, Chinese planting structure has not been adjusted accordingly, resulting in a large amount of grain being used as feed [5]. Therefore, improving the forage productivity has become a priority instead of a challenge.

Artificial grassland is an effective way to improve the pasture production in China. Intercropping is the cultivation of two or more crop species on the same land. Compared to the monocrop, intercropping systems help to increase total yield and yield stability of cropping systems, decrease the negative impacts of environmental problems, and promote the coordinated utilization of light energy, nutrients, water and other resources [6]. Intercropping systems play an important role in sustainable intensification of agriculture and could help improve the productivity of artificial grassland.

Many typical species have been used in intercropping systems, and maize/peanut intercropping systems could increase grain yield and soil available nutrients (avail P and



avail N) and change the soil microbial composition [7]. Agroforestry systems combining crops and pasture such as wheat/alfalfa have been promoted to improve the depth of 20–150 cm soil nitrogen fertility [8]. Legume-based intercropping systems such as switchgrass/milkvetch have been studied [9]. Previous studies focused on rhizodeposition transfer [10], water use efficiency [11], nutrients (such as phosphorus) [12,13], photosynthetically active radiation [14], microbial properties of rhizospheres [7] and the control of weeds, pests and wind erosion [15–17].

It has been reported that the yield of crops is determined by mixture of environment main effect (E), genotype main effect (G), and genotype environment interaction (GE), and E explains most (up to 80% or higher) of the total yield variation [18]. Thus, there are differences in the cultivation measures among varied environments. Oat–vetch mixture and the conventional tillage system might be the best options for rainfed farmers for improving fodder yield and quality [19,20]. Inner Mongolia Autonomous Region, with the maximum grassland area in China, has recently received more attention. Most areas of Inner Mongolia Autonomous Region are farming–pastoral zones and are under semiarid climatic conditions. The shortage of water resources is a main ecological problem. Oat (*Avena Sativa* L.) is high-quality forage, and it was considered to be tolerant to drought stress [21]. The oat planting area of Inner Mongolia Autonomous Region is the maximum in China. Promoting the development of the oat industry has economical and ecological value. Common vetch (*Vicia Sativa* L.) is a versatile pasture crop in leguminosae that provides multiple benefits for the farm, which is rich in protein, calcium, phosphorus and other nutrients, and is also resilient in regards to drought tolerance [22]. The common vetch plant is relatively low and conducive to growth and harvest [23]. Therefore, common vetch relies on oats as a climbing support, which is beneficial to increase its photosynthetic area, improve its light conditions, make full use of the space advantage and promote its forage yield and quality [24]. However, the suitable intercropping patterns of oat and common vetch in Inner Mongolia Autonomous Region have not yet been deeply explored; therefore, they have been studied in this experiment. In the present study, the following topics were addressed: (1) What are the effects of different oat/vetch intercropping treatments on yield and nutritional quality? and (2) Which is the best intercropping ratio?

## 2. Materials and Methods

### 2.1. Experimental Site

This study was carried out in the farming–pastoral zone of Wuchuan County in typical semiarid conditions, Inner Mongolia Autonomous Region, China (coordinates 41°12′ N and 111°42′ E, 1657 m above sea level). The region is a typical mid-temperate continental monsoon area, and the climate parameters are shown in Table 1. During the production seasons (May-August), the average annual sunshine duration was 8.75 h. The mean maximum air temperature was 28.8 °C, while the mean minimum air temperature was 8.2 °C. In each year, there was a 124d frost-free period, and the mean annual rainfall was 321.2 mm from 2015–2016 with at least 1807 mm evaporation. The average annual humidity is 26%. The soil is classified as Aridisol [25], with a pH of 7.8 (1:2.5 soil: deionized water), 15.32 g·kg$^{-1}$ soil organic matter, 0.88 g·kg$^{-1}$N, 0.42 g·kg$^{-1}$P, 17.45 g·kg$^{-1}$K, 61.07 mg·kg$^{-1}$ available N, 12.16 mg·kg$^{-1}$ available P (Olsen-P), 104.83 mg·kg$^{-1}$ available K in the 0–30cm layer.

### 2.2. Experimental Design

The experimental design was a randomized complete block with 3 replicates. The treatments were (1) vetch, (2) oats and five intercropping systems, including (3) oat intercropped with common vetch at seeding ratios 1:1, (4) at seeding ratio 1:2, (5) at seeding ratio 1:3, (6) at seeding ratio 2:1, and (7) 3:1. The size of each plot was 6 m wide by 6 m long. Oat cultivar VAO-2, selected by the Inner Mongolia Agricultural University Oats Industry Research Center, and common vetch cultivar Ximu 333/A was provided by the College of Agriculture, Hohhot, China. The row spacing of oats and common vetch was 20 cm in both

monoculture and intercrop systems. The overall plant seeding rate was 150 kg·hm$^{-2}$ oat and 60 kg·hm$^{-2}$ common vetch (Table 2). The based fertilizer is diammonium phosphate, 150 kg·hm$^{-2}$ (DAP:18-46-0). The sowing date of oat and common vetch was on 4th May in 2015, 1st May in 2016, and the corresponding dates of harvesting were 75, 90 and 105 DAS, respectively. Twice during the season sprinkle, irrigation was applied to the crop at a rate of 300 m$^3$·hm$^{-2}$. Weeds, pests, and diseases were controlled according to farmers' practice.

**Table 1.** Weather data for Wuchuan county, Inner Mongolia, China, in 2015 and 2016.

| Climate Factor | Year | Month | | | |
|---|---|---|---|---|---|
| | | 5 | 6 | 7 | 8 |
| T max (°C) | 2015 | 23.2 | 25.0 | 28.8 | 28.3 |
| | 2016 | 22.5 | 24.8 | 28.1 | 28.1 |
| T min (°C) | 2015 | 8.6 | 13.4 | 16.7 | 15.3 |
| | 2016 | 8.2 | 13.2 | 16.8 | 16.8 |
| Rainfall [a] (mm) | 2015 | 5.1 | 19.2 | 20.3 | 9.0 |
| | 2016 | 11.5 | 36.0 | 46.4 | 38.3 |
| Sunshine hours (h d$^{-1}$) | 2015 | 10.1 | 8.0 | 9.4 | 9.1 |
| | 2016 | 9.7 | 8.6 | 8.0 | 8.7 |

[a] Rainfall amounts are monthly totals; other values represent monthly averages of daily values.

**Table 2.** The experiment design.

| Treats | Code | Oat Seeding Rate (kg·hm$^{-2}$) | Common Vetch Seeding Rate (kg·hm$^{-2}$) |
|---|---|---|---|
| Oat sole | O | 150.0 | —— |
| Common vetch sole | V | —— | 60.0 |
| Oat + Common vetch 1:1 | 1:1 | 75.0 | 30.0 |
| Oat + Common vetch 1:2 | 1:2 | 50.0 | 40.0 |
| Oat + Common vetch 1:3 | 1:3 | 37.5 | 45.0 |
| Oat + Common vetch 2:1 | 2:1 | 100.0 | 20.0 |
| Oat + Common vetch 3:1 | 3:1 | 112.5 | 15.0 |

Note: O refers to oat, V refers to common vetch, the same for the following. The sowing amount of intercropping treatment was determined by monocrop. For example, OV1:1, which meant that the sowing amount of oats and vetch was half of that of monoculture, and OV1:2, which meant that the sowing amount of oats in intercropping was one-third of that of monoculture, and vetch in intercropping was two-thirds of that of monoculture. The same applies to OV1:3, OV2:1 and OV3:1.

### 2.3. Sampling and Measurements

To determine the yield, plants were hand-harvested at 3 cm above the soil surface three times (75 DAS, when oat heading and vetch flowering, 90 DAS corresponding to oat filling and vetch clamping and 105 DAS corresponding to oat milk ripe and vetch ripening); for each time, 1 m$^2$ plants were harvested. The cuttings were separated from weeds immediately and weighed for the fresh weight. Then, the sample was dried in an oven (ULM 800, Member GmbH, Schwa Bach, Germany) at 105 °C for 0.5 h, then dried at 80 °C to a constant weight to weigh the aboveground dry matter biomass. For the analysis of CP, CF, NDF and ADF, all the dried samples were smashed with a small pulverizer and passed through a 1 mm sieve to be analysed.

### 2.4. Nutrition Quality Analysis

Crude protein (CP) was analysed by the AOAC [26] method and calculated as N × 6.25. Crude fat (CF) was analysed by Soxhlet extraction method (GB/T5512). The CP yield and CF yield (kg·hm$^{-2}$) was calculated by multiply of crude protein, crude fat concentration and dry matter yield (kg·hm$^{-2}$), respectively. Fibre was measured as described by Van Soest et al. (Horwitz W. et al., 2007), with the samples being sequentially digested using an Ankom 220 Fiber Analyzer (Ankom Technology, Fairport, NY, USA) in accordance with the recommendations for neutral detergent fibre (NDF) and acid detergent fibre (ADF) analyses. The relative feed value (RFV) index was estimated as the digestible dry matter (DDM)

content of the samples based on ADF values. First, the dry matter intake (DMI) potential (as a percentage of body weight, BW) was calculated from NDF values, and the index was then calculated as DDM multiplied by DMI as a % of BW and divided by 1.29 [27].

$$DDM = 88.9 - (0.779 \times \%ADF)$$

$$DMI = 120/(\%NDF)$$

$$RFV = (DDM \times DMI)/1.29$$

*2.5. Statistical Analyses*

Statistical analyses were performed using SPSS version 20.0. The mean differences were compared using Duncan's multiple range t-test. Comparisons with $p < 0.05$ were considered significant. The relationships among forage aboveground biomass and quality were examined by performing Pearson correlation analysis by R (4.0.3) with the corrplot package. Origin software (Version 8.5; Northampton, MA, USA) was used to draw figures.

**3. Results**

*3.1. Green Grass Yield*

The shoot fresh weight (SFW) of common vetch and oat was influenced by growing period (Table 3). The fresh weights of aboveground at 90 DAS were significantly higher than that at 75 DAS ($p < 0.05$). The SFW was highest in the OV3:1 plot (expect at 105 DAS in 2015) and increased by 24.9%, 128.7% (at 75 DAS in 2015), 32.5%, 28.3% (at 90 DAS in 2015), 34.1%, 63.2% (at 75 DAS in 2016), 42.8%, 69.4% (at 90 DAS in 2016), and 12.2%, 45.8% (at 105 DAS in 2016) compared to both sole oat and sole common vetch, respectively. There was no significant difference between OV3:1 and OV2:1 at 75, 90 and 105 DAS in 2015 ($p > 0.05$).

**Table 3.** The effect of different intercropping ratios of oat and common vetch on fresh grass yield (t·hm$^{-2}$).

|  |  | **75 DAS** | **90 DAS** | **105 DAS** |
|---|---|---|---|---|
| | O | 24.00 Bb ± 0.63 | 28.12 Acd ± 1.06 | 27.46 Ac ± 1.13 |
| | V | 13.11 Bf ± 0.25 | 29.04 Ac ± 1.88 | 28.60 Aabc ± 0.70 |
| | OV1:1 | 22.43 Cc ± 0.75 | 32.99 Ab ± 1.37 | 27.94 Bbc ± 0.91 |
| 2015 | OV1:2 | 17.33 Bd ± 0.63 | 25.86 Ade ± 0.24 | 25.10 Ad ± 0.96 |
| | OV1:3 | 15.05 Be ± 1.45 | 23.56 Ae ± 2.49 | 24.11 Ad ± 1.06 |
| | OV2:1 | 29.46 Ba ± 0.38 | 35.22 Aab ± 0.16 | 30.56 Ba ± 1.19 |
| | OV3:1 | 29.98 Ba ± 1.31 | 37.27 Aa ± 0.68 | 29.66 Bab ± 1.4 |
| | O | 22.92 Cd ± 1.26 | 34.04 Ac ± 1.96 | 29.87 Bc ± 1.96 |
| | V | 18.83 Ce ± 1.18 | 28.69 Ae ± 0.93 | 22.99 Bd ± 1.18 |
| | OV1:1 | 22.58 Cd ± 0.93 | 31.61 Ad ± 0.54 | 28.83 Bc ± 1.16 |
| 2016 | OV1:2 | 27.72 Bbc ± 0.33 | 32.79 Acd ± 1.79 | 30.15 ABbc ± 1.63 |
| | OV1:3 | 26.11 Bc ± 1.32 | 34.59 Ac ± 1.13 | 32.43 Aab ± 1.10 |
| | OV2:1 | 28.81 Cb ± 0.56 | 37.32 Ab ± 0.20 | 33.33 Ba ± 1.23 |
| | OV3:1 | 30.73 Ca ± 0.98 | 48.60 Aa ± 1.35 | 33.51 Ba ± 1.16 |

Note: The data represent the mean ± SD; *n* = 3. Different capital letters indicate significant differences among different harvest days, and different lowercase letters indicate significant differences among different treatments ($p < 0.05$), the same as below.

*3.2. Dry Matter Biomass*

The dry matter yield showed an increasing trend with the improving growth period and was significantly affected by intercropping growth period (Table 4). In this intercropping research, the dry biomass of oat and common vetch were significantly affected by the seeding rate and different days after sowing ($p < 0.05$) (Table 5). For all planting arrangements, the increase of shoot dry matter (SDM) was clearly noticed at 105 days of the growth period compared to the 75- and 90-day growth periods. For the 105 DAS, under

the treatment of OV3:1, the SDM of mixed forage (both oat and common vetch) in 2015 was 19.91 t·hm$^{-2}$, increased by 23.2%, 59.8% compared to sole oat and common vetch, which indicates this intercropping pattern could contribute to improved forage yield. Under the treatment of OV3:1, the SDM of mixed forage (both oat and common vetch) in 2016 was 20.71 t·hm$^{-2}$ increased by 24.1% and 37.1% compared to sole oat and common vetch.

**Table 4.** The forage yield and quality of oat and common vetch in the sole and intercropped systems under different seedling periods in 2015 and 2016.

| Factors | Df | GY | DMB | CPC | CPY | CFC | CFY | NDF | ADF | RFV |
|---------|----|----|-----|-----|-----|-----|-----|-----|-----|-----|
| T | 54 | ** | ** | ** | ** | ** | ** | ** | ** | ** |
| D | 18 | ** | ** | ** | ** | ** | ** | ** | ** | ** |
| Y | 9 | ** | ** | ** | ** | NS | ** | NS | NS | * |
| T × Y | 54 | ** | ** | NS | ** | NS | ** | NS | NS | * |
| D × Y | 18 | ** | ** | ** | NS | * | NS | NS | NS | NS |
| T × D | 108 | ** | ** | ** | ** | NS | ** | NS | NS | NS |
| T × D × Y | 108 | ** | ** | ** | NS | NS | NS | NS | ** | ** |

* and ** denote significant difference at 0.05, 0.01 probability levels, NS means not significant difference. T, D and Y refer to treatment, days after sowing and year respectively.

**Table 5.** The effect of different intercropping ratios of oat and common vetch on dry matter biomass (t·hm$^{-2}$).

| Year | Treatment | 75 DAS | 90 DAS | 105 DAS |
|------|-----------|--------|--------|---------|
| | O | 4.19 Ca ± 0.33 | 6.87 Bb ± 0.69 | 16.16 Abc ± 0.68 |
| | V | 2.38 Cc ± 0.01 | 7.26 Bb ± 0.69 | 12.46 Ad ± 0.81 |
| | OV1:1 | 4.10 Ba ± 0.42 | 4.93 Bc ± 0.48 | 12.84 Ad ± 0.68 |
| 2015 | OV1:2 | 3.35 Cb ± 0.54 | 7.21 Bb ± 0.83 | 15.17 Ac ± 1.00 |
| | OV1:3 | 2.10 Cc ± 0.15 | 6.88 Bb ± 0.38 | 16.81 Ab ± 0.32 |
| | OV2:1 | 3.90 Cab ± 0.17 | 7.44 Bb ± 0.18 | 19.11 Aa ± 0.44 |
| | OV3:1 | 4.11 Ca ± 0.26 | 9.36 Ba ± 0.33 | 19.91 Aa ± 0.96 |
| | O | 4.96 Cc ± 0.48 | 7.12 Bc ± 0.15 | 16.69 Ac ± 0.14 |
| | V | 3.42 Ce ± 0.15 | 5.92 Bd ± 0.24 | 15.11 Ad ± 0.98 |
| | OV1:1 | 4.27 Cd ± 0.29 | 6.84 Bc ± 0.59 | 18.51 Ab ± 0.11 |
| 2016 | OV1:2 | 5.33 Cbc ± 0.27 | 7.48 Bc ± 0.25 | 18.21 Ab ± 0.62 |
| | OV1:3 | 5.89 Cab ± 0.42 | 7.48 Bc ± 0.55 | 18.06 Ab ± 0.71 |
| | OV2:1 | 6.51 Ca ± 0.45 | 9.64 Bb ± 0.64 | 19.27 Ab ± 0.70 |
| | OV3:1 | 6.37 Ca ± 0.25 | 12.61 Ba ± 0.18 | 20.71 Aa ± 0.90 |

### 3.3. Crude Protein Content

The crude protein contents in both oat and common vetch are shown in Figure 1. Crude protein concentrations from the harvested oat and common vetch were determined and expressed as percentages of dry mass (% DM) (Figure 1). CP contents were higher in sole common vetch than in sole oat. At 75, 90 and 105 DAS, the CP contents of sole common vetch were 15.5%, 13.7% and 13.8% in 2015, were 16.2%, 15.4% and 13.5% in 2016, and the monoculture common vetch showed the highest CP contents among all the treatments at 75 to 105 DAS. For the five intercrop arrangements, the CP contents were higher at 90 DAS than at 75 DAS to 105 DAS.

The overall range of CP contents under five intercropping patterns among three grown periods varied between 6.9% and 13.5% in 2015 while it changed between 8.6% and 15.9% in 2016. In 2015, the CP contents of OV1:1, OV2:1 at 90 DAS were higher than that of OV1:2, OV3:1, the CP contents of OV1:1 were 18.4% higher than sole oat. In 2016, the CP contents at 90 DAS of OV1:3 were 3.3%, 17.8% greater than sole common vetch and sole oat. The ANOVA analysis further indicated that as a result of 90 DAS, CP content suffered significantly more than 75 and 105 DAS (except sole common vetch), resulting in 2016 ($p < 0.05$).

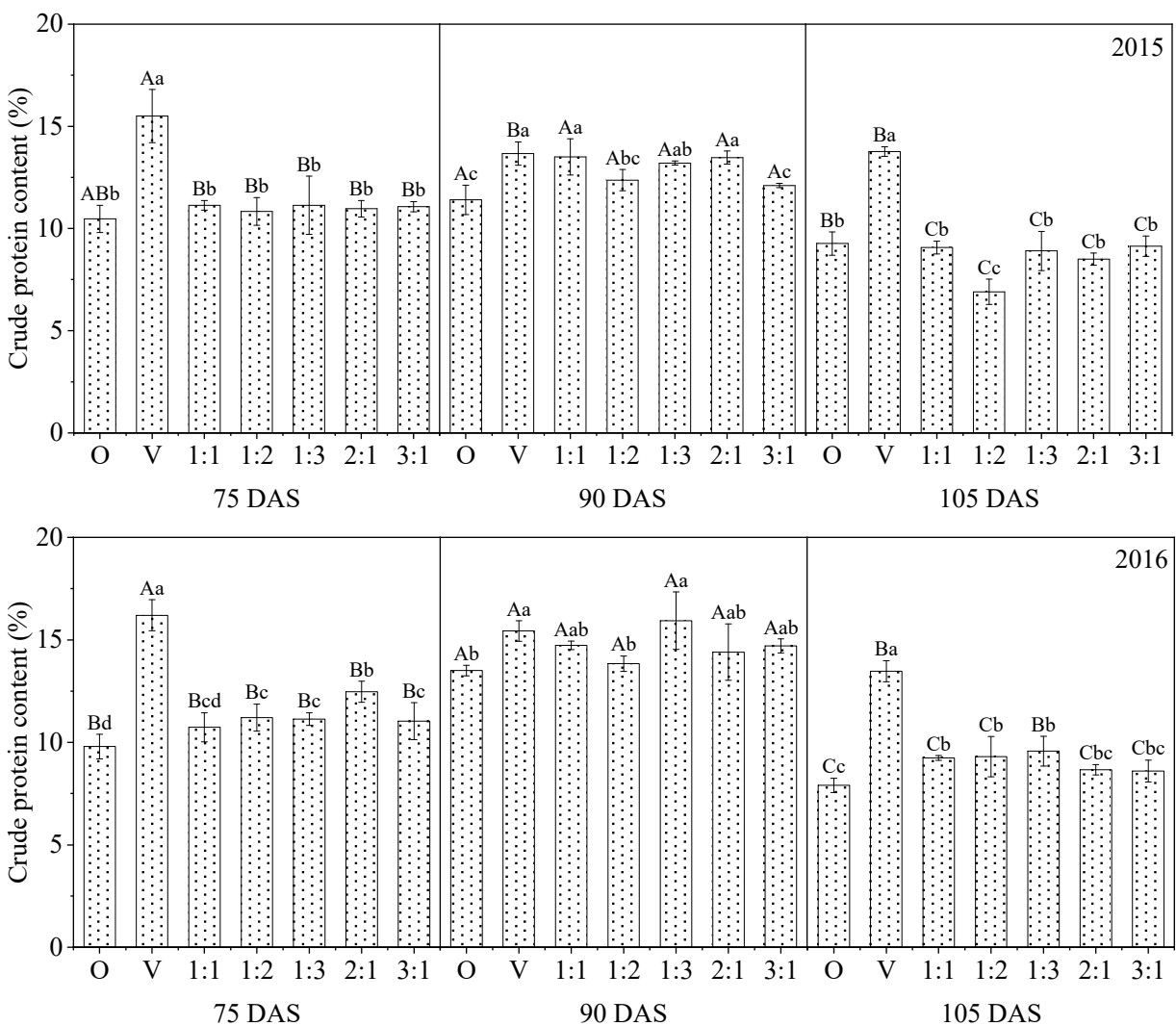

**Figure 1.** The effect of different intercropping ratio of oat and common vetch on crude protein content of forage. Note: Different capital letters indicate significant differences among different harvest days, and different lowercase letters indicate significant differences among different treatments (*p* < 0.05), the same as below.

### 3.4. Crude Protein Yield

The CP yield at 105 DAS were higher than at 75 and 90 DAS (*p* < 0.05). Among the five intercropping treatments, crude protein yield had the highest value in the OV3:1 at 90, 105 DAS (Figure 2). In 2016, OV3:1 at 105 DAS had the highest CP yield (1780.62 kg·hm$^{-2}$) among the intercropping treatments, followed by the mixture of oat with common vetch (2:1) (1670.52 kg·hm$^{-2}$) (*p* > 0.05), which were 26.7%, 35.1% higher than oat monoculture, respectively.

### 3.5. Crude Fat Content

The CF contents varied in response to intercropping treatments and growth period. Figure 3 shows the CF contents of both cultivars gradually rising, then decreasing with the growth period. Analysis of variance (ANOVA) indicated there was no significant difference (*p* > 0.05) for crude fat contents among different stages in 2015, a similar trend to that of CF content between 2015 and 2016. No significant difference was observed between 75 and 105 DAS growth period whatever the intercropping treatments (except OV3:1) were in 2016, OV3:1 processing crude fat reached maximum 7.9% at 90 DAS in 2016. It was 22.3% higher than monoculture oat at the same time. Compared to monoculture common vetch,

intercropping increased crude fat contents by 7.6–12.4%, 2.5–6.8%, 11.1–21.0% at 75, 90, 105 DAS respectively in 2015, and increased by 8.8–13.7%, 15.6–26.9%, 9.0–24.0% at 75, 90, 105 DAS, respectively, in 2016.

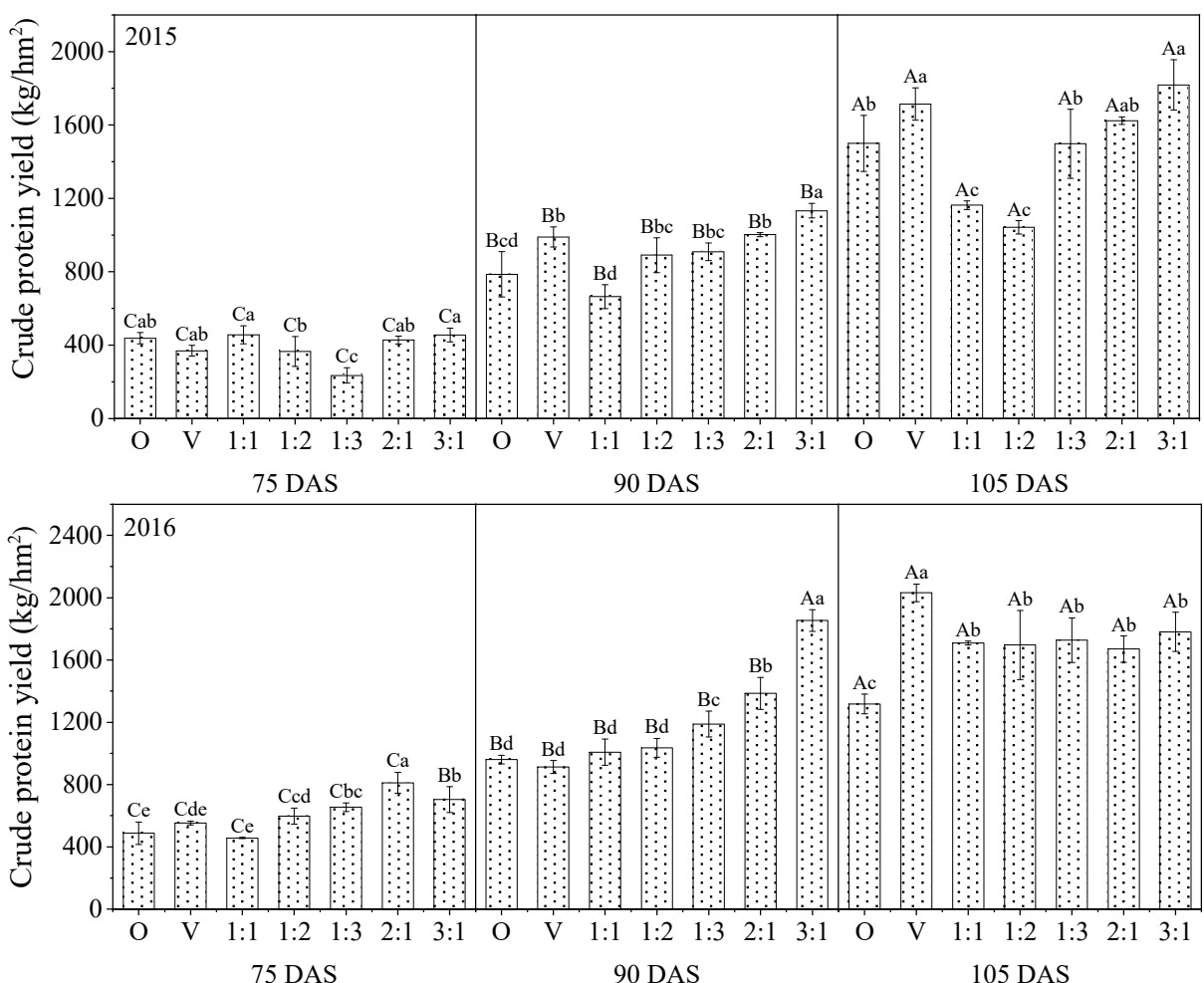

**Figure 2.** The effect of different intercropping ratios of oat and common vetch on crude protein yield of forage.

### 3.6. Crude Fat Yield

In this experiment, a similar trend to that of CP yield was observed for crude fat yield. It increased as the days after sowing increased (Figure 4). Crude fat yield analysis showed that OV3:1 was highest, at 284.53 kg·hm$^{-2}$ (75 DAS), 664.69 kg·hm$^{-2}$ (90 DAS) in 2015. There was a 21.7%, 73.2% and 28.8%, 55.8% increase in crude fat yield of OV3:1 mixture, respectively, as compared to sole oat and common vetch crude fat yield in 2015 and 2016. No differences between OV3:1 and OV2:1 (432.16 kg·hm$^{-2}$ and 449.88 kg·hm$^{-2}$ at 75 DAS in 2016, 1254.78 kg·hm$^{-2}$ and 1310.88 kg·hm$^{-2}$ at 105 DAS in 2016, respectively) were found (Figure 4).

### 3.7. Neutral and Acid Detergent Fiber

NDF and ADF are considered as important forage quality characteristics. The NDF, ADF, and RFV were not influenced by the interaction between intercropping and growth period (Table 4). Table 6 summarized the fiber content (NDF and ADF) of oat and common vetch at different growth stages. Interestingly, the amount of fiber (NDF and ADF) tended to decrease with advancing stages.

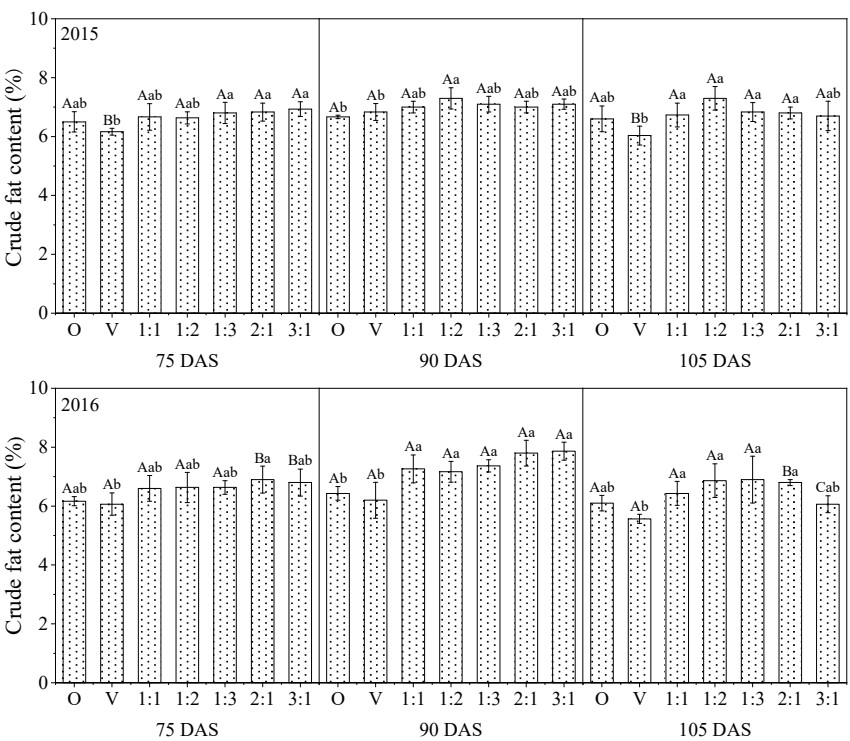

**Figure 3.** The effect of different intercropping ratio of oat and common vetch on crude fat content of forage.

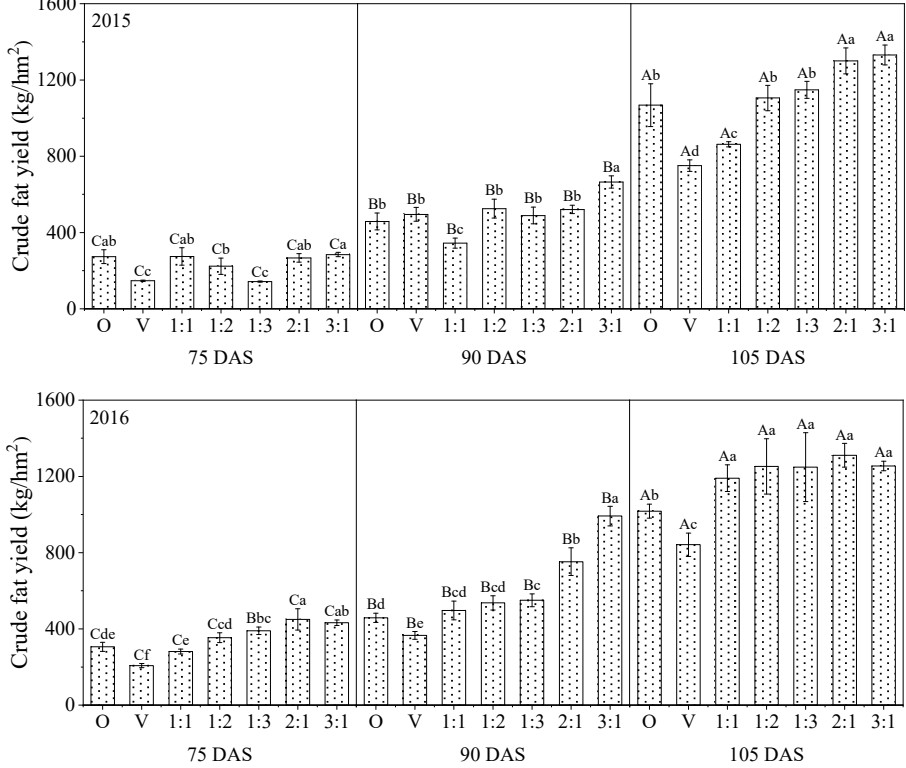

**Figure 4.** The effect of different intercropping ratios of oat and common vetch on crude fat yield of forage.

**Table 6.** Acid detergent fiber fraction and neutral detergent fiber fraction of oat and common vetch at different days after sowing in 2015.

| Year | Treatment | 75 DAS | | | 90 DAS | | | 105 DAS | | |
|---|---|---|---|---|---|---|---|---|---|---|
| | | NDF | ADF | RFV | NDF | ADF | RFV | NDF | ADF | RFV |
| 2015 | O | 66.00 Aa ± 2.00 | 40.00 Aab ± 2.00 | 81.39 Bc ± 0.27 | 54.67 Ba ± 3.06 | 32.00 Bab ± 2.00 | 109.14 Abc ± 7.87 | 52.67 Ba ± 1.15 | 32.67 ABab ± 5.77 | 112.17 Abc ± 9.43 |
| | V | 65.33 Aab ± 1.15 | 41.33 Aa ± 3.06 | 80.74 Bc ± 3.39 | 55.33 Ba ± 2.31 | 35.33 Aa ± 6.43 | 103.38 Ac ± 10.78 | 58.00 Ba ± 2.00 | 34.00 Aa ± 3.46 | 100.18 Ac ± 5.65 |
| | OV1:1 | 62.00 Aab ± 2.00 | 34.67 Aabc ± 6.11 | 92.88 Bbc ± 6.76 | 51.33 Aa ± 1.15 | 30.67 Aabc ± 1.15 | 117.86 Abc ± 3.78 | 58.67 Aa ± 10.07 | 33.33 Aab ± 4.62 | 101.73 ABc ± 17.46 |
| | OV1:2 | 64.67 Aab ± 7.02 | 35.33 Aabc ± 4.16 | 89.05 Bbc ± 11.85 | 50.00 Bab ± 3.46 | 29.33 ABabc ± 3.06 | 123.3 Abc ± 10.09 | 48.67 Ba ± 3.06 | 28.67 Babc ± 1.15 | 127.52 Aab ± 6.62 |
| | OV1:3 | 60.00 Aabc ± 3.46 | 35.33 Aabc ± 2.31 | 95.26 Bbc ± 2.80 | 49.33 Bab ± 5.03 | 26.67 Abc ± 4.16 | 129.73 Aab ± 19.3 | 53.33 ABa ± 6.11 | 31.33 Aabc ± 6.11 | 113.17 ABbc ± 11.98 |
| | OV2:1 | 58.67 Abc ± 2.31 | 32.00 Abc ± 6.93 | 101.66 Aab ± 11.15 | 48.67 Aab ± 7.02 | 26.00 Abc ± 2.00 | 133.32 Aab ± 22.03 | 50.67 Aa ± 4.62 | 25.33 Abc ± 4.16 | 127.56 Aab ± 11.24 |
| | OV3:1 | 53.33 Ac ± 4.16 | 31.13 Ac ± 4.76 | 113.39 Ba ± 13.16 | 42.67 Ab ± 2.31 | 24.67 ABc ± 2.31 | 152.16 Aa ± 7.27 | 48.00 Aa ± 8.72 | 24.00 Bc ± 2.00 | 138.52 ABa ± 20.37 |
| 2016 | O | 66.67 Aa ± 3.06 | 38.47 Aa ± 3.32 | 82.24 Bc ± 1.16 | 54.00 Bab ± 2.00 | 30.67 Ba ± 1.15 | 112.09 Abc ± 4.5 | 50.67 Ba ± 1.15 | 32.67 Bab ± 3.06 | 116.6 Aab ± 6.85 |
| | V | 66.27 Aa ± 0.46 | 39.33 Aa ± 2.31 | 81.78 Bc ± 2.30 | 56.00 Ba ± 3.46 | 32.00 Ba ± 3.46 | 106.44 Ac ± 5.81 | 55.33 Ba ± 6.43 | 36.00 ABa ± 4.00 | 103.64 Ab ± 17.63 |
| | OV1:1 | 61.33 Abc ± 1.15 | 37.33 Aa ± 2.31 | 90.76 Bab ± 3.89 | 51.33 Bab ± 1.15 | 28.67 Ba ± 1.15 | 120.69 Aab ± 3.84 | 52.00 Ba ± 6.93 | 33.33 ABab ± 4.16 | 114.3 Aab ± 19.43 |
| | OV1:2 | 63.53 Aab ± 0.81 | 38.00 Aa ± 3.46 | 86.81 Bbc ± 3.51 | 54.00 Ba ± 2.00 | 28.67 Ba ± 5.77 | 114.62 Abc ± 4.50 | 50.00 Ca ± 2.00 | 30.67 ABab ± 3.06 | 121.15 Aab ± 8.30 |
| | OV1:3 | 60.00 Abc ± 2.00 | 39.33 Aa ± 2.31 | 90.44 Bab ± 5.56 | 52.67 Bab ± 2.31 | 30.00 Ba ± 2.00 | 115.97 Abc ± 7.78 | 54.67 ABa ± 4.62 | 29.33 Bab ± 4.16 | 112.7 Aab ± 6.85 |
| | OV2:1 | 61.77 Abc ± 3.06 | 36.00 Aa ± 2.00 | 91.80 Bab ± 5.04 | 51.33 Bab ± 4.16 | 27.33 Ba ± 4.62 | 123.12 Aab ± 12.78 | 49.33 Ba ± 4.16 | 28.67 ABb ± 4.16 | 125.81 Aab ± 6.00 |
| | OV3:1 | 59.33 Ac ± 2.31 | 36.67 Aa ± 1.15 | 94.73 Ba ± 4.99 | 49.33 Bb ± 1.15 | 26.00 Ba ± 2.00 | 129.53 Aa ± 5.84 | 46.67 Ba ± 2.31 | 29.33 Bab ± 3.06 | 131.85 Aa ± 7.35 |

OV3:1 at 90 DAS in 2015 accumulated the lowest NDF contents among all the treatments and decreased by 22.0%, 22.9% compared to the sole oat and sole common vetch. At 75 and 90 DAS, the highest NDF content occurred in the monoculture. OV3:1 at 105 DAS in 2015 displayed the lowest ADF contents, and decreased by 26.5%, 29.4% compared to the sole oat and sole common vetch. The monoculture treatment generated the highest ADF contents compared to other arrangements at 75 DAS in 2015 and 2016. The contents of ADF and NDF at 90 DAS were significantly lower than in 75 DAS in 2016 ($p < 0.05$).

As NDF and ADF decrease, there is an increase in RFV. RFV was higher in OV3:1 than in other intercropping treatments, monoculture oat and monoculture common vetch in 2015–2016, showing a significant difference. The RFV of all intercrops ranged from 86.81 (in OV1:2 at 75 DAS, 2016) to 152.16 (in OV3:1 at 90 DAS, 2015) across the two years.

*3.8. Correlation Analysis*

The Pearson's correlation coefficients between forage quality and biomass were estimated (Figure 5). The results showed that dry matter was significantly related to CFY, RFV and CPY. Similarly, GM was significantly related to CFY, RFV, DM and CPY. However, CFY was only significantly related to CPC, ADF and NDF, and RFV was significantly affected by ADF and NDF.

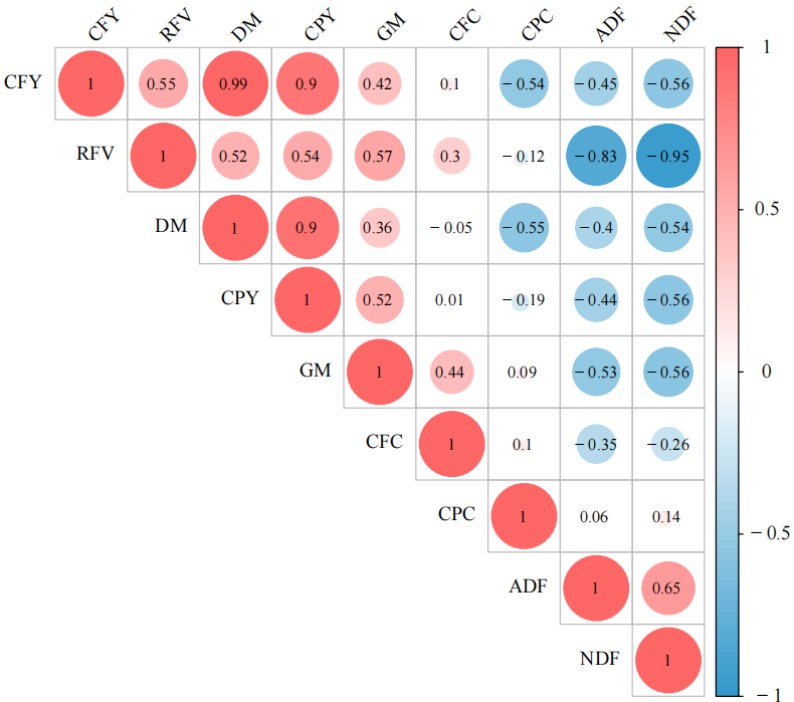

**Figure 5.** The relationship between forage quality and biomass. Green biomass (GM), Dry matter (DM), Crude protein content (CPC), Crude protein yield (CPY), Crude fat content (CFC), Crude fat yield (CFY), Acid detergent fibre (ADF), Neutral detergent fibre (NDF), Relative feed value (RFV).

## 4. Discussion

Inner Mongolia province is a typical agro-pastoral ecotone, which is under limited water resources. The oat planting area of Inner Mongolia is the largest in China, and oat is one of the most important forages in the local husbandry system. Vetch is a drought-tolerant legume with appreciable economic benefits, which could be potentially combined with oat in intercropping systems. It has been well documented that the forage production varied greatly with different growing environments. Harvesting stages significantly affected dry matter yield (DMY) [28]. In Inner Mongolia, a suitable oat/vetch intercropping system has not been successfully developed.

Significant advantages of legume/cereal intercropping systems were observed over other systems in terms of crop yield [29,30]. For example, higher yields are reported when pea is intercropped with barley [31]. Li et al. reported that in common vetch-oat row intercropping systems, the aboveground biomass was increased compared with sole cropping in Qinghai–Tibetan plateau [32]. Our results were consistent with Li et al. In the present study, the oat/vetch intercrop experiment was set in Inner Mongolia Autonomous region. Yield is the key index to reflect the performance of mixed-seeded forage. Among the 7 intercrop treatments, OV3:1 increased the biomass yield compared to the monoculture at the growth period 105d. One reason for this may be that in grass-legume mixtures, legumes can provide nutrients for grass through the soil food web [33] to stimulate plant growth, thus increasing yield. Another reason may be that the high percentage of oats in mixed pastures promoted the production. In addition, the biomass yield was lower in OV1:1 than in OV2:1 and OV3:1, and this may be attributed to the shading environment for vetch led by the taller plant height of oat. In the shading environment, the fraction of intercepted PAR, photosynthesis for common vetch decreased. This phenomenon indicated that biomass yield could be affected by the row proportion design in intercrop system. Zhang et al. found no significant yield advantage for oats/vetch intercropping in their study, and this may be attributed to the row design [34].

Crude protein content is regarded as important criteria for forage quality evaluation [35]. For sole cropping, the crude protein contents in vetch were higher than in oat in this study. Of all the growing conditions of this experiment, the highest CP yield was found at 105 days after sowing, followed by 90 and 75 DAS. This was because DM yield at 105 DAS increased CP yield. The highest crude protein content was observed in the OV1:1, followed by the OV2:1 at 90 DAS in 2015. A similar result was reported by A.S. Lithourgidis [35], who found that the barley/wheat intercrop system could increase the CP contents compared to sole crops. CP yield is the index which combined DM yield and CP content, and in this study the maximum CP yield obtained in OV3:1 at 105 DAS in 2015, and increased by 21.3%, 6.1% compared to monocropped oat and monocropped common vetch, indicated this treatment could help facilitate the improvement of dry matter and crude protein contents.

Fatemeh (2019) reported that intercrops had higher protein content where the seeding ratios of common vetch increased in the intercrop [36]. Legumes play an important role in improving forage quality in intercropping [37,38]. The optimal row configurations RI (one row of common vetch alternated with one row of oat) and S32 (three rows of common vetch alternated with two rows of oat), which displayed the highest RUE, LER, yield, can be applied in common vetch-oat strip intercropping for sustainable forage production in alpine regions [20,39]. This result was consistent with AMONGE et al. 2013, who found that the crude protein contents were increased by oat-linseed intercropping [20]. Crude fat could provide energy for livestock, and in this study, crude fat contents in vetch were greatly enhanced by oat-vetch intercrop (such as OV1:3) compared to sole crop treatment. The acid detergent fiber is another important quality index for forage. In the present study, the ADF contents were decreased in intercrops compared to sole crops, which is beneficial for livestock digestion. These results indicate that a key strategy to improve forage quality is to adopt a suitable intercropping system. However, there are conflicting results; for example, no differences in ADF composition between vetch-oat intercrops and sole crop were observed [40]. This may be attributed to the varied environments.

In this study, negative correlations between SDM and protein were observed, and this is consistent with the results of [41]. The treatment OV3:1 could facilitate the improvement of SDM and protein contents at the same time compared to the sole oat, suggesting that a suitable intercrop pattern could facilitate the increase of yield production and quality contents.

A previous study found that the dry matter, growth rate and plant height were affected by growth period in the rice/maize intercrop system, which is consistent with our results. We found the shoot dry matter increased under 105 DAS was higher than under 90 and

75 DAS, while the crude protein contents under 105 DAS were lower than under 75 DAS. With the advance of the growth period, the content of CP showed a downward trend. It may be that the CP content of oats in the milk stage is lower than that in the filling stage due to the shedding of oats in the wax ripening stage. In addition, with the postponement of the growth stage, the contents of NDF and ADF also showed a downward trend; that is, the yield of NDF and ADF in the milk stage was lower than that in the filling stage. In particular, NDF decreased significantly in the later stage. It may be that with the maturity of oat seed, the proportion of its weight in the whole plant continues to increase, while the ratio of NDF to ADF decreases. This phenomenon suggested that growth period affected yield and quality, which should receive more attention.

## 5. Conclusions

Forage yield and quality varied in responding to five oat-common vetch intercrop patterns in the farming–pastoral zone of Inner Mongolia. Among all the treatments, OV3:1 at 105 DAS achieved the highest shoot dry matter, increased by 23.2%, 59.8% in 2015 and increased by 24.1%, 37.1% in 2016, compared to sole oat and sole common vetch. CP yield increased with intercropping OV3:1 compared to the sole oat. The highest CF contents were observed in OV3:1 at 90 DAS in 2016. The lowest NDF and ADF were observed in OV 3:1. Growth period affected forage quality and yield, which was demonstrated by the higher SDM and CF yield in 105 DAS compared to 75, 90 DAS, and the higher CP contents in 90 DAS compared to 75, 105 DAS.

To determine the best harvest time of forage grass, many factors need to be considered, such as forage species or varieties, the law of growth and development of forage grass, external environmental conditions, the growth and utilization of regenerated grass and so on. Only by comprehensively considering the above aspects can the best harvest time be determined.

**Author Contributions:** J.Q., J.B. and L.L. conceived and designed the experiments; Y.Z. and Q.C. performed the experiments; G.C. performed the statistical analysis; J.Q. and L.L. wrote the paper. All authors have read and agreed to the published version of the manuscript.

**Funding:** This study was funded by Inner Mongolia science and Technology major special project "Research on rational Utilization technology and Integration mode of water Resources in dryland Area" (2020ZD0005-0401) and the National Natural Science Foundation of China (31901459).

**Institutional Review Board Statement:** Not applicable.

**Acknowledgments:** We would like to thank the Oat research Team for field and data collection.

**Conflicts of Interest:** The authors declare no conflict of interest.

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
