# Peer review of "Influence of Different Proportion Intercropping on Oat and Common Vetch Yields and Nutritional Composition at Different Growth Stages"

_agronomy, doi:10.3390/agronomy12081908_

Round 1

Reviewer 1 Report

Overall, this appears to be relevant research conducted specific to a region in China. Other similar research in China has been conducted and this add to that data. The focus of the paper and introduction could be improved especially in relation to work that has already been completed on forage oats mixed with legumes. This is an area of study that has been important to other countries/regions as well. Generally when I think of intercropping I am drawn to the row crop literature or grain crops or vegetables. It is common in forages to mix species to obtain the benefit of legumes both for nitrogen as well as forage quality. 

The research question needs to be clarified as there is harvest date as a treatment and included in the analysis but does not seem to be represented in the statistical design. The statistical analysis needs to be more clear.

The methods could also use some work as it is unclear how the seeding ratios were determined and if actually planted in rows (each species) how that was accomplished. 

The results again are confusing because there is a harvest date treatment included but not sure if this is a split plot design. It would be good to understand what harvest dates equate to growth stages of the crop. 

Discussion is focused on parameters of yield and quality that are well known to change as forages become more mature. This is not a unique finding however understanding how the ratio of legume to oat might impact that finding is important. This should be really drawn out more in the discussion. I don't believe the correlations add much to this paper and should be removed.

Other comments below.

Sole vs Mono crop – generally see referred to as monocrop meaning one.

Line 10: environment and cultivation practices that vary according to growing environment

Line 11: An oat-common vetch

Line 14: patters sole oat, and sole vetch

Line 17: (oat intercropped with common vetch

Line 18: consistently use 3 significant digits 24.1%

Line37: seems like the better word here is priority versus challenge.

Line 39: artificial grassland? I have never heard this term before. Do you mean alternatives to perennial grasslands such as annuals? Is the goal to improve pasture or to increase or enhance the amount of pasture in the diet?

Line 76: What is intercropping mode?

Line 93: The experimental design was a randomized complete block with 3 replicates.

Line 94: The treatments were vetch, oats, and five intercropping systems including

Line 101: It says overall seeding rate – does this mean in the monocultures? When reviewing Table 2 and the seed rates for each component it was unclear how the amount of seed was derived for each ratio. This needs to be explained in the text.

Line 104 & 105: Talks about harvest dates – was each plot harvest 3 times or is harvest date a subplot within the main plot?

Line 105&106: Twice during the season sprinkle irrigation was applied to the crop at a rate of xxx

Line 112: Is the harvest from the same area in the plot each time? or is it considered a subplot within the main plot? This would impact the analysis or study design.

Line 117: What type of pulverizer or mill did it pass through a 1-mm sieve? All of the sample?

Line 120: use abbreviation for CP and CF

Line 135-138: What were considered the treatments crop ration and harvest time? What was the actual research question and how will this experiment answer the question? Is the question what combination of oats and vetch provide the most yield and quality OR When is the best time to harvest these combinations to achieve optimum yield/quality OR both? Also was the analysis by year or across years? Was there a year by treatment interaction indicating that years should be analyzed separately?

Need clarification on statistical analysis and experimental design.

Line 141: would make more sense for the heading to say forage yield or treatment yields…the treatments were not all green grass

Line 142: growing period – do you mean harvest date?

Table 4: are these dry matter yields? Table 5 is actual DM  yields. Now I am curious why both are reported. Dry matter basis is the standard.

Line 169-170: Table has all ** so denotes significant at 0.01 but also includes in the footnote 0.05 with * - can remove this

Table 4 comes before Table 3 in the text – Tables should be numbered as they appear in the text.

Correlation table does not add much to the data analysis. The increase in RFV should be expected as NDF and ADF decrease. It is well documented that forage quality declines as forages become more mature.

Line 293: Refers to the ratios of 3 rows oats and 1 row of vetch. This further confuses the seeding ratios. How were single rows of each crop planted is this how the ratios were determined and also setup in the field. Were they hand planted?

There is numerous papers evaluating oat as a forage and also vetch. Generally forages such as oats are harvested based on growth stage to optimize yield and quality. There was no reference to forage data and it is well known/documented that harvest in the boot to soft dough stage changes quality and yield of oat (and other forages) quite considerably. It would be good to know what growth stage oats/vetch were harvested at and compare to other data of this nature. In the discussion it suggests that the oats were harvested in the milk stage and beyond but this is the only place where this reference is made.

Overall abbreviations not consistently used in the text.

Reviewer 2 Report

Manuscript "Influence of different proportion intercropping on Oat and Common Vetch yields and nutritional composition at different growth stages" is very interesting.

General comments:

Authors analysed effects of different oat/vetch intercropping treatments on yield and nutritional quality.
Authors selected the best intercropping mode.

Detailed comments:

Description of "2.5. Statistical analyses" is very poor. Lack information about distribution of observed traits. Lack is onformation about analysis of correlations.
Authors used three factors: year, treatment and days after seeding. Lack is results of three-way analysis of variance.

My suggestion:

Line 19: "crude protein (CP)" not "CP (Crude protein)"
Table 4: Mean values and standrard deviations (SD) or standard error (SE)???
Table 4: letters after mean values not SD or SE.
Table 5: letters after mean values not SD or SE.
Line 195: "analysis of variance (ANOVA)" not "ANOVA".
Line 199: "90 days" not "90days".
Line 201: "Fig. 2" not "Fig.2".
Table 6: letters after mean values not SD or SE.

Paper needs major revision.

Round 2

Reviewer 2 Report

Few corrections are necessary:

Table 4: letters should be after mean values not SD.
Table 5: letters should be after mean values not SD.
Table 6: letters should be after mean values not SD.

Paper needs minor revision.